# Assessment Tools of Biopsychosocial Frailty Dimensions in Community-Dwelling Older Adults: A Narrative Review

**DOI:** 10.3390/ijerph192316050

**Published:** 2022-11-30

**Authors:** Vincenzo De Luca, Grazia Daniela Femminella, Roberta Patalano, Valeria Formosa, Grazia Lorusso, Cristiano Rivetta, Federica Di Lullo, Lorenzo Mercurio, Teresa Rea, Elena Salvatore, Nilufer Korkmaz Yaylagul, Joao Apostolo, Rosa Carla Silva, Carina Dantas, Willeke H. van Staalduinen, Giuseppe Liotta, Guido Iaccarino, Maria Triassi, Maddalena Illario

**Affiliations:** 1Dipartimento di Sanità Pubblica, Università degli Studi di Napoli Federico II, 80131 Napoli, Italy; 2Dipartimento di Scienze Mediche Traslazionali, Università degli Studi di Napoli Federico II, 80131 Napoli, Italy; 3Dipartimento di Medicina Clinica e Chirurgia, Università degli Studi di Napoli Federico II, 80131 Napoli, Italy; 4Specializzazione in Igiene e Medicina Preventiva, Università degli Studi di Roma Tor Vergata, 00133 Roma, Italy; 5Dipartimento di Scienze Biomediche Avanzate, Università degli Studi di Napoli Federico II, 80131 Napoli, Italy; 6Faculty of Health Science, Akdeniz University, 07070 Antalya, Turkey; 7Health Sciences Research Unit: Nursing (UICISA:E), Nursing School of Coimbra (ESEnfC), Avenida Bissaya Barreto, 3004-011 Coimbra, Portugal; 8SHINE 2Europe, 3030-163 Coimbra, Portugal; 9AFEdemy—Academy on Age-Friendly Environments in Europe, 2806 ED Gouda, The Netherlands; 10Dipartimento di Biomedicina e Prevenzione, Università degli Studi di Roma Tor Vergata, 00133 Roma, Italy

**Keywords:** frailty, active and healthy ageing, frailty prevention, screening tools, community-dwelling older adults, digital health, integrated care, frailty risk factors

## Abstract

Frailty is a complex interplay between several factors, including physiological changes in ageing, multimorbidities, malnutrition, living environment, genetics, and lifestyle. Early screening for frailty risk factors in community-dwelling older people allows for preventive interventions on the clinical and social determinants of frailty, which allows adverse events to be avoided. By conducting a narrative review of the literature employing the International Narrative Systematic Assessment tool, the authors aimed to develop an updated framework for the main measurement tools to assess frailty risks in older adults, paying attention to use in the community and primary care settings. This search focused on the biopsychosocial domains of frailty that are covered in the SUNFRAIL tool. The study selected 178 reviews (polypharmacy: 20; nutrition: 13; physical activity: 74; medical visits: 0; falls: 39; cognitive decline: 12; loneliness: 15; social support: 5; economic constraints: 0) published between January 2010 and December 2021. Within the selected reviews, 123 assessment tools were identified (polypharmacy: 15; nutrition: 15; physical activity: 25; medical visits: 0; falls: 26; cognitive decline: 18; loneliness: 9; social support: 15; economic constraints: 0). The narrative review allowed us to evaluate assessment tools of frailty domains to be adopted for multidimensional health promotion and prevention interventions in community and primary care.

## 1. Introduction

The demographic changes taking place in Europe and worldwide will have profound implications in the planning and delivery of social and health services, due to the concomitant increase in the complexity of the health needs of the older adult population [1]. The COVID-19 pandemic has been imposing health challenges of unknown dimensions, both for the increased risk of an adverse outcome in the case of infection in older adults with multimorbidity [2] and for the deterioration in their quality of life as a result of loneliness and impaired access to health and social services. Mortality from COVID-19 has been increasing especially among vulnerable subgroups of populations [3], meaning that frail older adults are exposed to an increased risk of adverse outcomes. Although it is not an inevitable condition of aging, frailty is characterized particularly in those over 65 years old by a dynamic state of vulnerability with weakness and reduction in the physiological reserve which entails an increased risk of lower quality of life, falling, institutionalization, disability, and death [4]. The prevalence of frailty is age related [5] and is therefore linked to the demographic development patterns of the population, with different implications in terms of health needs that must be managed differently depending on the time of onset and the socioeconomic context of individuals. The older adult at risk of frailty presents losses in one or more domains of human functioning (physical, psychological, social) caused by the influence of a number of variables and which increase the risk of adverse outcomes [6]. A proactive and integrated approach to frailty prevention is therefore essential, strengthening the interaction between all professionals involved in health and the social system, and also engaging informal caregivers.

The involvement of multiple stakeholders of the European Innovation Partnership on Active and Healthy Aging (EIP on AHA) in the A3 Action Group [7] allowed researchers to identify and implement innovative approaches to the prevention and management of frailty, such as the one proposed by the SUNFRAIL European project (Grant n. 664291), integrating the biomedical paradigm of frailty [8] with the “biopsychosocial” paradigm [9]. This allowed researchers to take into account all influences of the individual health trajectory towards frailty, namely environmental, physical, educational, socioeconomic, and psychological factors. The SUNFRAIL project produced, as a result, a tool for the early identification of older adults at risk of frailty in different settings [10], and aims to develop, validate, and test an innovative model of intervention to improve the prevention and treatment of frailty and the management of multimorbidity [11,12,13]. The screening tool, which consists of only nine items, can be used by General Practitioners or other health service professionals and community actors, such as Community Nurses. In the case of risk, SUNFRAIL items can be connected to other in-depth tools for the assessment of specific dimensions (Figure 1). The SUNFRAIL tool aims to identify frailty risk factors in the older adult population earlier to generate alerts orienting subsequent diagnostic assessments for health promotion, disease prevention, and other targeted interventions. SUNFRAIL allows health professionals to carry out a very early screening based on the first contact of the older adult with primary care services, communities, and specialists to guide him/her to provide subsequent insights concerning the identified frailty domains (physical, neuropsychological, social). The SUNFRAIL questions were selected from tools used internationally: specifically, the Edmonton Frailty Scale [14], the Tilburg Frailty Indicator [15], and the Gerontopole Frailty Screening Tool [16].

The early diagnosis of frailty and the identification of its risk factors can enable health professionals to prevent or delay the occurrence of the adverse outcomes through targeted and timely interventions [11]. The SUNFRAIL is the first gateway between patients/beneficiaries and formal/informal services. It supports professionals to identify risk factors of frailty in apparently robust individuals. The objective of this narrative review is to identify assessment tools, to be coherently connected to each of the SUNFRAIL tool items, and further evaluate the corresponding dimensions, in case of risk in one or more biopsychosocial frailty domains. Such a more in-depth assessment is intended to develop targeted intervention strategies for frailty prevention dedicated to community-dwelling older persons, acting on the dimension identified as being at risk or compromised.

Indeed, the identification of further scales for each of the SUNFRAIL domain brings the potential of providing a dataset on the over 65 population, which is useful for the implementation of personalized prevention and health promotion interventions for the independent and healthy living of community-dwelling older adults, based on an in-depth risk factors assessment. The next step is to link specific domains with further clinical and diagnostic evaluations to allow for the coherent integration of innovative approaches and health interventions of care and cure addressing the compromised domains [17].

## 2. Materials and Methods

This review uses the International Narrative Systematic Assessment (INSA) [18] tool for narrative reviews as a method. This narrative review uses the following Population Implementation Comparator Outcome Study (PICOS) approach:Population: older people, aged 65 years or older;Implementation/indicator: frailty domains tool(s) (polypharmacy, weight loss, physical activity, medical visits, falls, memory loss, loneliness, support network, economic constraints);Comparator: n/a;Outcome: multidimensional frailty screening or frailty prevention in community-dwelling older adults;Study: review, systematic review, and meta-analysis.

The review selection was performed using the Embase and PubMed databases. A side search was performed to identify other relevant articles, especially in those domains where the search did not yield satisfactory results.

The narrative review focused on the 9 domains of frailty that are covered in the SUNFRAIL screening tool. The search strings considered are included in Table 1.

The search was restricted to reviews published between January 2010 and December 2021 and was limited to the English language.

The inclusion criteria of this narrative review were frailty domains measurement in older people, aged 65 years or over.

The following papers were excluded: articles focused on the definition and models of frailty; observational, cross-sectional, or randomized controlled trials; and papers focused on a specific topic or illness or adverse outcome.

The review articles on the domains of frailty that did not include a comparison of the assessment tools used in the literature were excluded on the basis of the full text.

In step 2, an independent reviewer listed the available assessment tools for each item on the y-axis and compared the tools according to the needs of health systems for IT-supported frailty screening intervention in primary and community care on the x-axis. Each tool was evaluated according to the match against the following needs:used on older adults living at home;validation through a peer-reviewed study;number of items, as completeness of questions and time required for administration;specificity as the ability of the instrument to clearly distinguish healthy subjects from those in which the specific domain is impaired. This was assessed by the reviewer by analyzing the nature of the questions (qualitative/quantitative) and the presence or absence of a final score.;ease of use as the need for specific preparation in order to administer the instrument;usability by different professionals (doctors, nurses, social workers, etc.).

The reviewer scored each match against the need through a three-point Likert scale (1 being “barely addresses the need”; 2 being “partially addresses the need”; 3 being “fully addresses the need”).

## 3. Results

### 3.1. Polypharmacy

The process of the review paper selection is summarized in the flowchart of Figure 2.

In total, we extracted 15 tools for the assessment of prescription appropriateness and adherence in the older adult population (Appendix A).

The tools we most commonly found were the screening tool of older people’s prescriptions and the screening tool to alert the right treatment (STOPP/START) [19] (n = 17 review papers) followed by the Beers Criteria [20] (n = 12), Medication Appropriateness Index (MAI) [21] (n = 4), and Fit fOR The Aged list (FORTA) [22] (n = 4). These tools, as well as STOPPFrail [23], the Norwegian General Practice criteria (NORGEP) [24], the (EU)(7)-PIM list [25], the PRISCUS list [26], the Systematic Tool to Reduce Inappropriate Prescribing (STRIP) [27], Good Palliative–Geriatric Practice (GP-GP) [28], the Individualized Medication Assessment and Planning program (IMAP) [29], and the Zhan Criteria [30] are the main criteria used to review drug therapy in light of identified iterations. They are used by professionals, mainly physicians and pharmacists, to identify potential drug interactions, which may have a negative impact on the health of the patient. Other tools, such as the Drug Regimen Unassisted Grading Scale (DRUGS) [31], Medication Management Ability Assessment (MMAA) [32], and Self-Efficacy for Appropriate Medication Use Scale (SEAMS) [33] are used to assess the older adult’s ability to adhere to prescriptions and to independently take care of themselves. Table 2 shows the prescription appropriateness and adherence assessment tools scores.

### 3.2. Weight Loss

We included for review 13 papers (Figure 3) and we extracted 15 tools aimed to assess malnutrition in older adults (Appendix A).

The best-rated nutrition screening tools are the PREDIMED [34] and the 10-point Mediterranean diet scale [35]. They are easy to use and measure adherence to the Mediterranean diet. The Mini Nutritional Assessment (MNA) [36], including the Mini Nutrition Assessment Short Form (MNA-SF) [37], is a more reliable tool for nutritional screening, but needs assessments that cannot be performed by nonmedical professionals. The Nutritional Form for the Elderly (NUFFE) [38] is a self-assessment questionnaire developed for the hospital setting, but it takes into account all aspects considered important for nutritional assessment. The Malnutrition Universal Screening Tool (MUST) [39] is a very thorough and complex tool to submit. The Canadian Nutrition Screening Tool (CNST) [40], the Chapman Nutritional screening [41], and the Malnutrition Screening Tool (MST) [42] measure the nutritional status of the patient in the hospital setting. The DETERMINE Checklist [43] is not specific. The Seniors in the Community Risk Evaluation for Eating and Nutrition (SCREEN I) or (SCREEN II) questionnaire [44] places more emphasis on clinical judgement.

The Council on Nutrition Appetite Questionnaire (CNAQ) [42] and the Simplified Nutritional Appetite Questionnaire (SNAQ) [45] are not specific, as they measure appetite level. The SNAQ65+ [46] combines self-rated questions and anthropometric measurements. The SCALES (Sadness, Cholesterol, Albumin, Loss of weight, Eating, Shopping) questionnaire [47] measures weight loss associated with other risk factors and requires specific clinical expertise. Table 3 shows malnutrition assessment tools scores.

### 3.3. Physical Activity

The study selected 74 papers for review (Figure 4). In total, 25 tools aimed to assess physical activity in older adults were identified (Appendix A).

The tool that was found most often was the Short Physical Performance Battery (SPPB) (n = 40) [49]. It assesses the older adult’s ability to walk, get up from the chair, and remain balanced. Other instruments such as the five-chair stand test [50], Timed Up and Go Test [51] (n = 30), Four Square Step test [52], and Alternate Step Test (ATS) [53] measure several aspects of physical function individually. The SPPB is also used to assess fall risk. The 6-Minute Walking Test (6MWT) (n = 33) [54] and 5-Meter Walking Test (5MWT) [55] instruments are easy to use in both hospital and community settings to assess the older adult’s ability to walk. The Incremental Shuttle Walk Test (ISWT) [56] aims to simulate a cardiopulmonary exercise test using a field walking test. There have been several self-report questionnaires found to quantify the amount of daily physical activity of older adults. The widely used measures are the Physical Activity Scale for the Elderly (PASE) (n = 8) [57], Community Healthy Activities Model Program for Seniors (CHAMPS) questionnaire [58], Zutphen Physical Activity Questionnaire [59], Physical Activity and Sedentary Behavior Questionnaire (PASB-Q) [60], EPIC Physical Activity Questionnaire (EPAQ2) [61], International Physical Activity Questionnaire (IPAQ) [62], General Practice Physical Activity Questionnaire (GPPAQ) [63], and Longitudinal Aging Study Amsterdam Physical Activity Questionnaire (LAPAQ) [64], which aim to assess the duration, frequency, exertion level, and amount of physical activity undertaken over a seven-day period by individuals 65 years and older. The Stanford Brief Activity Survey [65], Women’s Health Initiative physical activity questionnaire (WHI-PAQ) [66], Physical Activity Scale for Individuals with Physical Disabilities (PASIPD) [67], and Duke Activity Status Index [68] assess recreational activities or exercises (mild, moderate, and strenuous) as well as household and yard activities.

The SF-36 (physical component scale) [69] measures self-perceived physical function and role limitations, including physical, bodily pain. The WOMAC physical function subscale [70] is a widely employed patient-reported measurement instrument for difficulty in physical function due to pain and discomfort in knee osteoarthritis patients. The Tinetti Performance-Oriented Mobility Assessment (POMA) [71], Mini Motor Test (MMT) [72], and Elderly Mobility Scale [73] evaluate physical performance by means of gait, balance, and stand-up tasks. Table 4 shows physical activity assessment tool scores.

### 3.4. Adherence to Medical Visits

The scientific production on the topic of adherence to medical visits is not extensive. The search only concerned reviews and meta-analyses on this specific topic, so it did not produce any results (Figure 5).

However, the analysis of the eligible papers highlighted several studies that have predominantly used checklists or single oral questions. Among the excluded papers, there is a study in which an eight-item tool was used to collect information on older adults’ access to primary care, continuity, and connectedness, which were considered to be enabling factors for the use of the emergency room [74].

### 3.5. Falls

The narrative review for fall items included 39 papers for review (Figure 6), through which 26 tools aimed to assess fall, and the risk of falling in older adults was identified (Appendix A).

The Falls Efficacy Scale (FES) [75] and its revised versions, the Modified Falls Efficacy Scale (MFES) [76] and the Short Falls Efficacy Scale—International (FES-I) [77] are the most common instruments for assessing confidence in older adults’ ability to perform Activities of Daily Living (ADL) without falling.

The Survey of Activities and Fear of Falling in the Elderly (SAFE) [78], Mobility Efficacy Scale (MES) [79], and University of Illinois at Chicago Fear of Falling Measure (UIC FFM) [80] measure the self-reported fear of falling. The Falls Risk Assessment Questionnaire (FRAQ) [81] aims to assess awareness and perception of falls risk.

The Fall Risk Index (FRI-21) [82] has been used to detect older adults’ decline in basic activities of daily living (BADL). The Fracture Risk Assessment Tool (FRAX) [83] estimates the risk of having a hip or other major fracture.

Tinetti POMA [71], Berg Balance Scale (BBS) [84], and Fullerton Advanced Balance (FAB) Scale [85] assess static and dynamic balance and fall risk in adults.

The Activities-specific Balance Confidence (ABS) scale [86] is a questionnaire developed to assess older individual’s self-perceived balance confidence in performing daily activities.

The Fall risk assessment tool (FRAT) [87], Morse Fall Scale [88], St. Thomas Risk Assessment Tool in Falling Elderly Inpatients (STRATIFY) [89], Conley Scale [90], Johns Hopkins Fall Risk Assessment Tool [91], Hendrich II Fall Risk Model tool [92], Austin Health Falls Risk Screening Tool (AHFRST) [93], and Falls Risk For Hospitalized Older People (FRHOP) [94] identify the risk factors of falls in hospitalized patients.

Falls Risk for Older People in the Community (FROP-Com) [95] and the Downton Fall Risk Index (DFRI) [96] aim to predict postdischarge injuries, identifying people who require further assessment and management. The Elderly Fall Screening Test (EFST) [97] aims to stratify the community-dwelling older adults population into a low and high risk of fall based on the fall history and observations of walking speed and gait style. The Home-Screen Scale [98] and Safety House Checklist [99] aim to identify environmental safety hazards in older adults’ homes. Table 5 shows the fall assessment tools scores.

### 3.6. Memory Loss

Regarding memory loss items, the study selected 12 papers for review (Figure 7). In total, 18 tools that aim to assess cognitive function in older adults were identified (Appendix A).

Our research showed that the most used tool is the Mini Mental State Examination (MMSE) [100] and its modified version (3MS) [101], which are questionnaires that healthcare professionals commonly use to check for problems with thinking, communication, understanding, and memory. The Montreal Cognitive Assessment (MoCA) [102] was designed as a tool for the rapid screening of mild cognitive impairment. The Cognitive Abilities Screening Instrument (CASI) [103], with a 0 to 100 score range, assesses several cognitive domains: attention and concentration, executive functions, memory, language, visual constructive skills, abstraction, calculation, and orientation. The short portable mental status questionnaire (SPMSQ or Pfeiffer) [104] aims to define the presence and intensity (mild, moderate, and severe) of cognitive disturbances of organic origin in elderly patients. The Abbreviated Mental Test (AMT) [105] is a quick-to-use screening test in general hospital usage. The 6-item cognitive impairment test (6-CIT) [106] is a useful dementia screening tool in primary care. The Clifton Assessment Procedures for the Elderly (CAPE) [107] evaluates the presence and severity of impairment in mental and behavioral functioning in long-term psychiatric patients. The Clinical Dementia Rating (CDR) [108] is used to quantify the severity of dementia using a semistructured interview. The Controlled Oral Word Association Test (COWAT) [109] and Isaacs Set Test (IST) [110] are neuropsychological measures of verbal fluency. The Trail Making Test A & B (TMT) [111] assesses spatial planning ability in a visual–motor task. The Rey Auditory Verbal Learning Test (RAVLT) [112] measures the immediate memory span and provides an assessment of learning. The clock-drawing test (CDT) [113] is a quick screening test for cognitive dysfunction, but it lacks sensitivity for the diagnosis of early or mild dementia. The Brief Cognitive Screening Battery (BCSB) [114] includes the MMSE, Verbal Fluency, clock-drawing, and Figure Memory Tests. The Mini-Cog [115] consists of a three-item recall test for memory and a simply scored clock-drawing test. The Telephone Interview for Cognitive Status (TICS) [116] is designed to be administered over the telephone. The Community Screening Interview for Dementia (CSID) [117] consists of a cognitive test and an interview on performance in everyday living. Table 6 shows the cognitive decline assessment tool scores.

### 3.7. Loneliness

We included for review 15 papers (Figure 8) and we extracted nine tools aimed to assess loneliness in older adults (Appendix A).

The UCLA Loneliness Scale [118] and the De Jong Gierveld scale [119] are the most common tools to measure subjective feelings of loneliness as well as feelings of social isolation in older adults. The Three-Item Loneliness Scale [120] is an interviewer-administered questionnaire developed from the Revised UCLA Loneliness Scale. The Questionnaire to define Social Frailty Status (QSFS) [121] aims to evaluate the state of social frailty through simple questions on daily social activity, social role, and social relationships. The Social Frailty Index (SFI) [122] assesses loneliness through self-reported survey questionnaires on topics such as living alone, having no education, the absence of a confident, infrequent contact, infrequent social activities, financial difficulty, and socioeconomic depravation. The Steptoe Social Isolation Index [123] measures family status, monthly contacts with children, family and friends, and participation in social groups. The 11-item Duke Social Support Index [124] includes subscales for social interaction and subjective support. Table 7 shows the loneliness assessment tool scores.

### 3.8. Social Support

We included for review five papers (Figure 9) and we extracted 15 tools to assess social support networks (Appendix A).

The UCLA Loneliness Scale [118] and the De Jong Gierveld scale [119] are also very frequently used to assess older adults’ support network and the presence of people who can take care of him/her in case of need. The Older Americans Resources and Services Program (OARS) and the Multidimensional Functional Assessment of Older Adults (MFAQ) [125] allows researchers to stratify the older adult population according to levels of social resources. The Inventory of Social Supportive Behaviors (ISSB) [126] assesses how often individuals received various forms of assistance during the preceding month. The Social Provision Scale [127] contains four subscales for each of the four social dispositions: attachment, social integration, material help, and value reassurance. The social support behaviors (SS-B) scale [128] and Medical Outcome Study Social Support Survey (MOS-SSS) [129] assess supportive behavior available from family and from friends in terms of emotional support, socializing, practical assistance, financial assistance, and advice/guidance. The Duke Social Support Index [124] assesses older adults’ social interaction and their satisfaction. The Multidimensional Scale of Perceived Social Support [130] and the Lubben Social Network Scale [131] measure the perceived adequacy of social support from three sources: family, friends, and significant others. The Berkman–Syme Social Network Index [132] allows researchers to categorize individuals into four levels of social connection: socially isolated, moderately isolated, moderately integrated, and socially integrated. The Personal Resource Questionnaire (PRQ) [133] provides descriptive information about the person’s resources and measures the respondent’s level of perceived social support. The six social support deficits [134] is a checklist including living alone, seeing a relative less often than once a week, lack of reciprocity with neighbors, lack of reciprocity between extended family members, relationship difficulty with one or more relatives, and dissatisfaction with support from family. The 2-Way Social Support Scale [135] assesses the emotional and instrumental support given and received. The Philadelphia Geriatric Center Morale Scale (PGCMS) [136] lonely dissatisfaction subscale measures the older person’s acceptance or dissatisfaction with the amount of social interaction they are presently experiencing. Table 8 shows support network assessment tool scores.

### 3.9. Economic Conditions

The search for reviews, systematic reviews, and meta-analyses on the assessment of the economic conditions of community-dwelling older people did not yield any results (Figure 10). The reason is that this domain belongs to a research field more related to the social sciences than to medicine and healthcare service organizations.

The side-search allowed us to identify two tools that can be used to assess economic conditions of older adults. Tarasenko and Schoenberg, 2017 [138], adopted a self-perceived income sufficiency questionnaire to assess resources in an underserved population. The Subjective Status Scale of Adler et al., 2000 [139], measures self-rated socioeconomic status by means of a 10-step scale whose extremes represent people who are better off—those who have more money, more education, and the most respected jobs—and those who are worse off—people who have less money, less education, and fewer or no respected jobs.

## 4. Discussion

In order to be effectively used for frailty screening and the implementation of prevention interventions on community-dwelling older adults, the assessment tools to be connected to SUNFRAIL should be aimed at the identification of risk factors that may lead to the impairment of one or more of the domains of frailty. The limitation of the search for reviews, systematic reviews, and meta-analyses did not allow for the inclusion of the most recent instruments in the results. As in the case of the Quick Mild Cognitive Impairment (QMCI) Screen for the assessment of mild cognitive decline, it is particularly useful for dementia prevention interventions [140]. As a consequence of the COVID-19 pandemic, many health systems are implementing a Family and Community Nurse (FCN) to meet the twin challenges of an ageing population and the increasing incidence of chronic conditions in order to prevent adverse events and enable healthcare services to encourage and ensure health improvement and wellbeing, instead of focusing exclusively on treating illness [141]. In the case of medication review support tools for older adults in polypharmacy, the assessment tools that are often used require specialist expertise and therefore cannot be administered by an FCN. Therefore, screening and prevention interventions should focus on adherence to drug therapies, just as assessment tools should measure the older adult’s ability to take care of themselves and manage their disease.

## 5. Conclusions

This narrative review made it possible to identify tools for the assessment of biopsychosocial frailty dimensions in community-dwelling older adults. These tools, appropriately selected and integrated, can represent the dataset for the implementation of targeted and timely health promotion, prevention, education, and information services in primary care. Further studies are needed to develop and validate a new service model for frailty screening in community-dwelling older adults by means of an IT tool that could support the healthcare providers by linking the elements of the SUNFRAIL tool to additional scales aimed at assessing impaired domains, thereby enabling professionals to develop appropriate intervention strategies.

## Figures and Tables

**Figure 1 ijerph-19-16050-f001:**
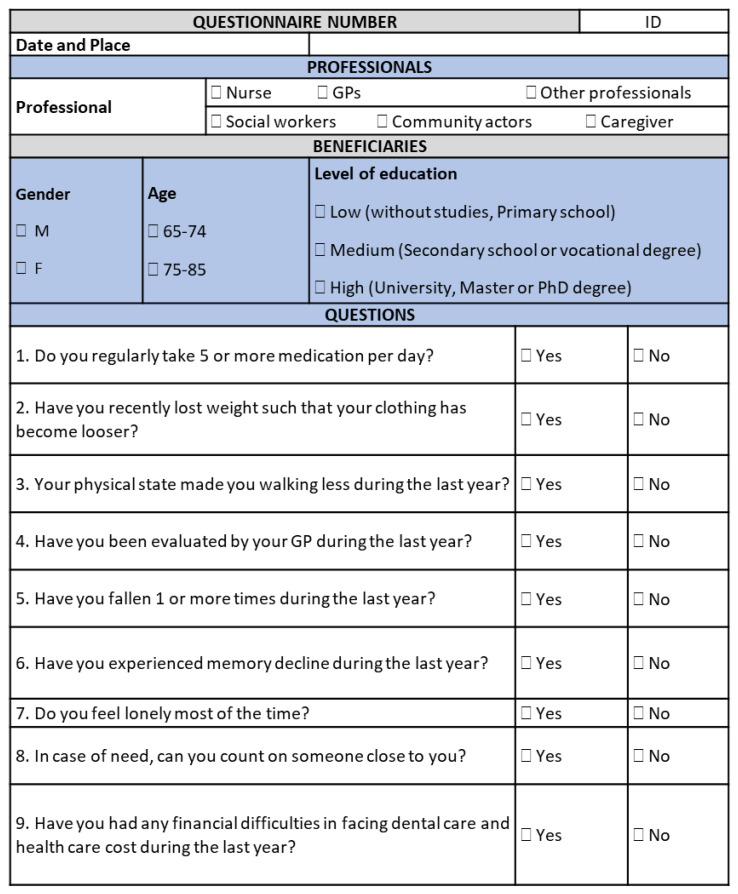
The SUNFRAIL screening tool.

**Figure 2 ijerph-19-16050-f002:**
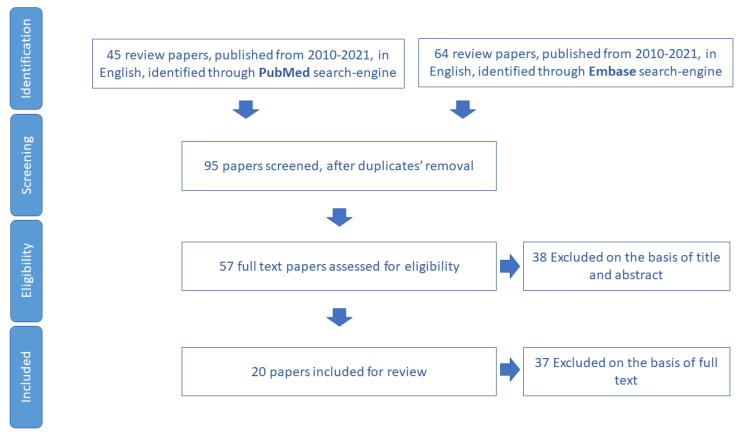
Process of study selection for polypharmacy domain.

**Figure 3 ijerph-19-16050-f003:**
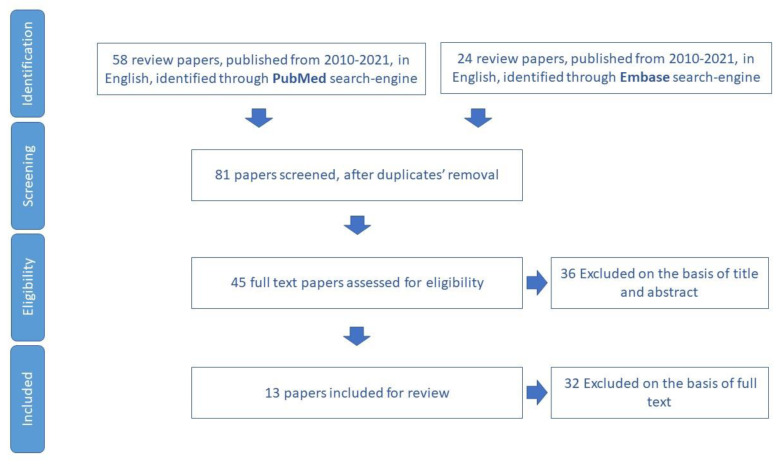
Process of study selection for nutrition domain.

**Figure 4 ijerph-19-16050-f004:**
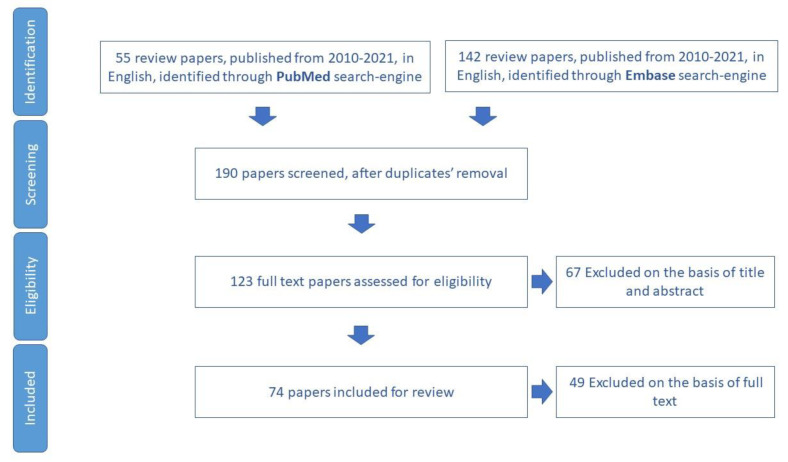
Process of study selection for physical activity domain.

**Figure 5 ijerph-19-16050-f005:**
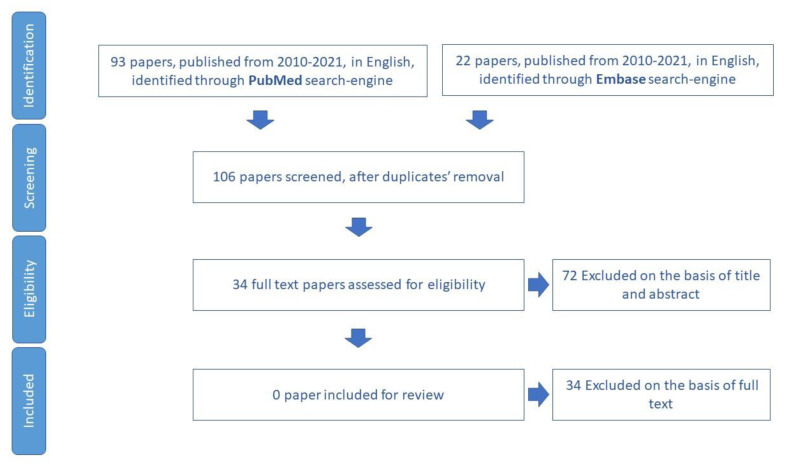
Process of study selection for assessment of adherence to medical visits.

**Figure 6 ijerph-19-16050-f006:**
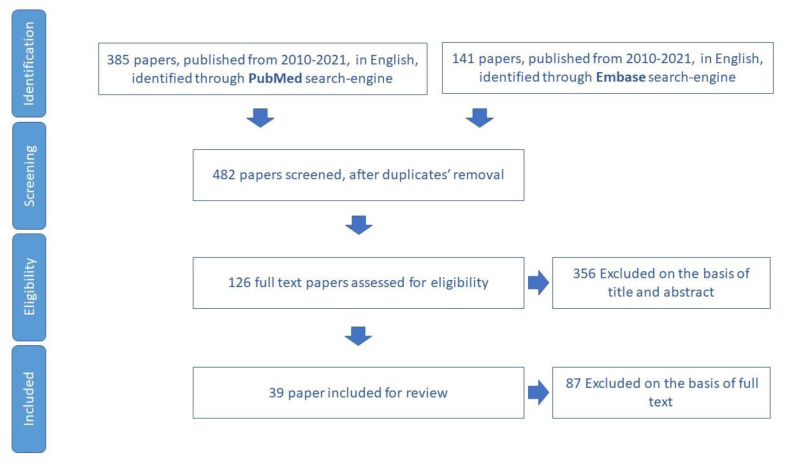
Process of study selection for fall risk assessment.

**Figure 7 ijerph-19-16050-f007:**
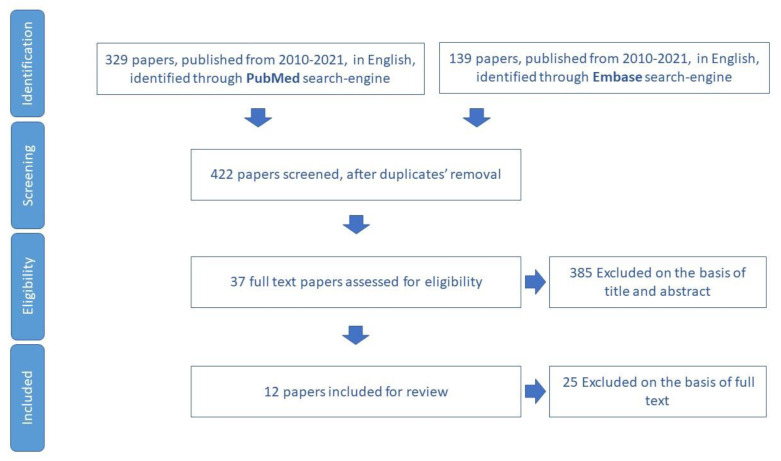
Process of study selection for the assessment of cognitive decline.

**Figure 8 ijerph-19-16050-f008:**
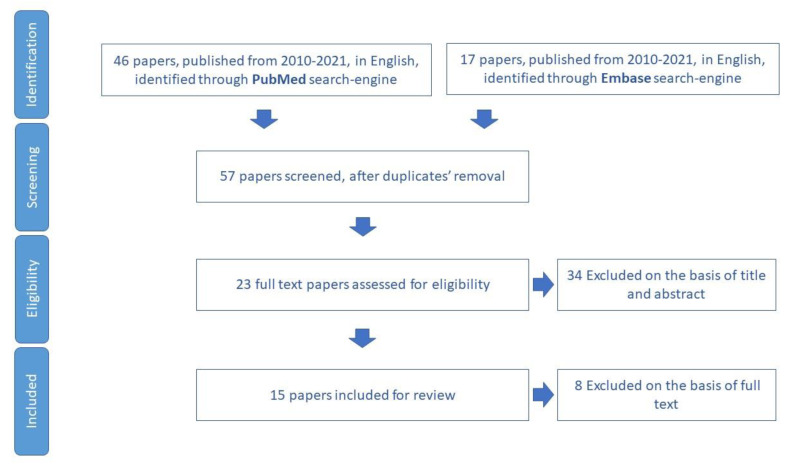
Process of study selection for the assessment of loneliness.

**Figure 9 ijerph-19-16050-f009:**
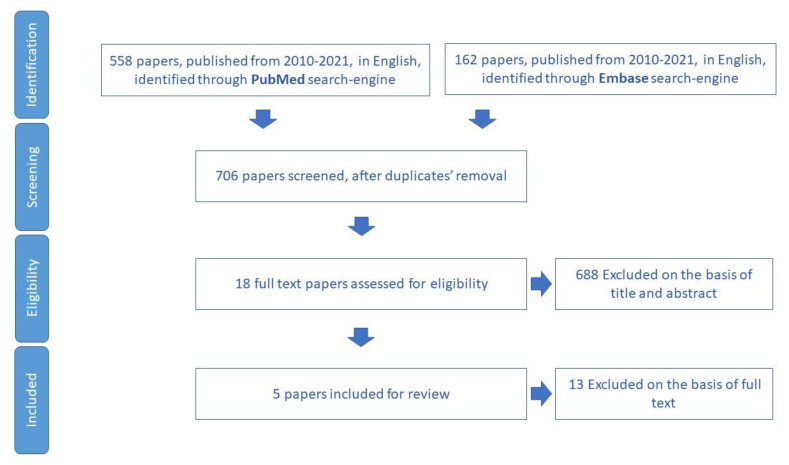
Process of study selection for social support domain.

**Figure 10 ijerph-19-16050-f010:**
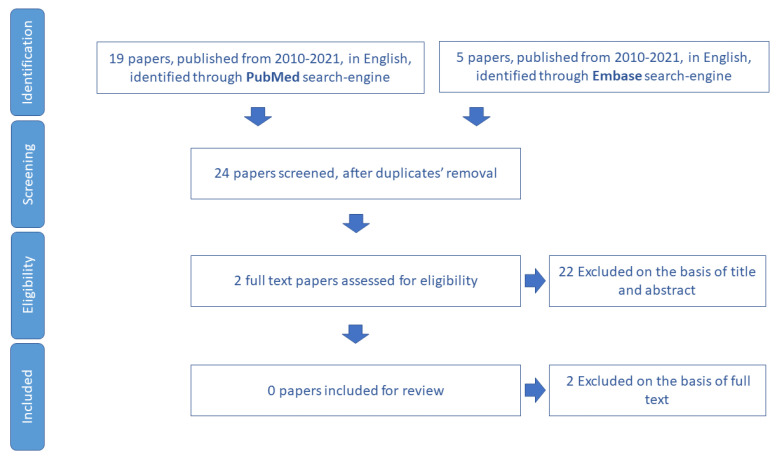
Process of study selection for assessment of economic constraints.

**Table 1 ijerph-19-16050-t001:** Search strings based on the SUNFRAIL’s frailty domain.

SUNFRAIL Frailty Domains	Search Strings
Polypharmacy	(frail elderly OR frailty OR elderly population) AND (tool OR screening tool OR assessment tool OR instrument OR test OR outcome measurement OR questionnaire) AND (review OR systematic review) AND (Polypharmacy)
Weight loss	(frail elderly OR frailty OR elderly population) AND (tool OR screening tool OR assessment tool OR instrument OR test OR outcome measurement OR questionnaire) AND (review OR systematic review) AND (weight loss) AND (nutrition)
Physical activity	(frail elderly OR frailty OR elderly population) AND (tool OR screening tool OR assessment tool OR instrument OR test OR outcome measurement OR questionnaire) AND (review OR systematic review) AND (physical activity) AND (walking)
Medical visits	(frail elderly OR frailty OR elderly population) AND (tool OR screening tool OR assessment tool OR instrument OR test OR outcome measurement OR questionnaire) AND (review OR systematic review) AND (medical visits)
Falls	(frail elderly OR frailty OR elderly population) AND (tool OR screening tool OR assessment tool OR instrument OR test OR outcome measurement OR questionnaire) AND (review OR systematic review) AND (falls)
Memory loss	(frail elderly OR frailty OR elderly population) AND (tool OR screening tool OR assessment tool OR instrument OR test OR outcome measurement OR questionnaire) AND (review OR systematic review) AND (cognitive decline)
Loneliness	(frail elderly OR frailty OR elderly population) AND (tool OR screening tool OR assessment tool OR instrument OR test OR outcome measurement OR questionnaire) AND (review OR systematic review) AND (loneliness)
Support network	(frail elderly OR frailty OR elderly population) AND (tool OR screening tool OR assessment tool OR instrument OR test OR outcome measurement OR questionnaire) AND (review OR systematic review) AND (support network)
Economic constraints	(frail elderly OR frailty OR elderly population) AND (tool OR screening tool OR assessment tool OR instrument OR test OR outcome measurement OR questionnaire) AND (review OR systematic review) AND (economic constraints)

**Table 2 ijerph-19-16050-t002:** Prescription appropriateness and adherence assessment tools scores.

	Use on Older Adults Living at Home	Validation	Number of Items	Specificity	Ease of Use	Usability by Different Professionals
STOPP/START [19]	2	3	2	3	1	1
Beers Criteria [20]	2	3	2	3	1	1
Medication Appropriateness Index (MAI) [21]	2	3	3	3	3	1
Fit fOR The Aged list (FORTA) [22]	2	3	1	3	1	1
STOPPFrail [23]	2	3	2	3	1	1
Norwegian General Practice criteria (NORGEP) [24]	2	3	2	3	1	1
(EU)(7)-PIM list [25]	2	3	2	3	1	1
PRISCUS list [26]	2	3	2	3	1	1
Systematic Tool to Reduce Inappropriate Prescribing (STRIP) [27]	3	2	3	1	3	2
Good Palliative–Geriatric Practice (GP-GP) [28]	1	1	2	1	2	1
Individualized Medication Assessment and Planning program (IMAP) [29]	3	3	2	2	1	1
Zhan Criteria [30]	2	3	2	3	1	1
Drug Regimen Unassisted Grading Scale (DRUGS) [31]	1	3	2	2	2	2
Medication Management Ability Assessment (MMAA) [32]	2	3	2	2	2	2
Self-Efficacy for Appropriate Medication Use Scale (SEAMS) [33]	1	3	2	2	2	2

The score assigned to each correspondence measures how well the instrument meets the needs of the study: 1 being “barely addresses the need”; 2 being “partially addresses the need”; 3 being “fully addresses the need”.

**Table 3 ijerph-19-16050-t003:** Malnutrition assessment tools scores.

	Used on Older Adults Living at Home	Validation	Number of Items	Specificity	Ease of Use	Usability by Different Professionals
PREDIMED [34]	3	3	3	3	3	3
10-point Mediterranean-diet scale [35]	3	3	3	3	3	3
Mini Nutritional Assessment (MNA) [36]	3	3	3	3	3	2
Mini Nutrition Assessment Short Form (MNA-SF) [37]	3	3	3	3	3	2
Nutritional Form for the Elderly (NUFFE) [38]	3	3	3	3	3	2
Malnutrition Universal Screening Tool (MUST) [39]	3	3	2	3	1	1
Canadian Nutrition Screening Tool (CNST) [40]	1	3	2	3	3	3
Chapman Screening tool [41]	1	2	2	1	2	2
Malnutrition Screening Tool (MST) [48]	1	3	2	2	3	3
DETERMINE Checklist [43]	2	3	1	1	1	1
Seniors in the Community Risk Evaluation for Eating and Nutrition questionnaire (SCREEN I) or (SCREEN II) [44]	3	3	3	3	1	1
Council on Nutrition Appetite Questionnaire (CNAQ) [42]	3	3	3	1	3	3
Simplified Nutritional Appetite Questionnaire (SNAQ) [45]	3	3	3	1	3	2
Short Nutritional Assessment Questionnaire (SNAQ65+) [46]	3	3	1	1	3	3
SCALES (Sadness, Cholesterol, Albumin, Loss of weight, Eating, Shopping) questionnaire [47]	3	2	2	1	1	1

The score assigned to each correspondence measures how well the instrument meets the needs of the study: 1 being “barely addresses the need”; 2 being “partially addresses the need”; 3 being “fully addresses the need”.

**Table 4 ijerph-19-16050-t004:** Physical activity assessment tools scores.

	Used on Older Adults Living at Home	Validation	Number of Items	Specificity	Ease of Use	Usability by Different Professionals
Short Physical Performance Battery (SPPB) [49]	3	3	2	3	3	3
5CST, five-chair stand test [50]	3	3	1	1	2	2
Timed Up and Go Test [51]	3	3	1	1	2	2
Four-Square Step test [52]	2	1	3	3	3	3
Alternate Step Test (ATS) [53]	3	3	1	1	2	2
6-Minute Walking Test (6MWT) [54]	3	3	3	3	2	2
5-Meter Walking Test (5MWT) [55]	2	3	3	2	3	3
Incremental Shuttle Walk Test (ISWT) [56]	2	2	2	1	1	1
Physical Activity Scale for the Elderly (PASE) [57]	2	3	3	2	3	3
Community Healthy Activities Model Program for Seniors (CHAMPS) [58]	3	3	1	3	1	1
Zutphen Physical Activity Questionnaire [59]	3	1	1	1	3	3
Physical Activity and Sedentary Behavior Questionnaire (PASB-Q) [60]	3	1	2	1	3	3
EPIC Physical Activity Questionnaire (EPAQ2) [61]	3	3	2	1	2	3
International Physical Activity Questionnaire (IPAQ) [62]	1	3	2	1	3	3
General Practice Physical Activity Questionnaire (GPPAQ) [63]	3	2	1	1	3	3
Longitudinal Aging Study Amsterdam Physical Activity Questionnaire (LAPAQ) [64]	3	3	3	2	2	2
Stanford Brief Activity Survey (SBAS) [65]	2	3	1	1	3	3
Women’s Health Initiative physical activity questionnaire (WHI-PAQ) [66]	1	3	2	2	2	2
Physical Activity Scale for Individuals with Physical Disabilities (PASIPD) [67]	2	2	1	1	2	3
Duke Activity Status Index [68]	3	2	3	2	2	3
SF-36 (physical component summary scale) [69]	1	3	1	1	1	2
WOMAC physical function subscale [70]	2	1	2	2	2	2
Tinetti Performance-Oriented Mobility Assessment (POMA) [71]	1	3	1	3	1	1
Mini Motor Test (MMT) [72]	3	2	1	3	1	2
Elderly Mobility Scale [73]	1	2	2	2	2	1

The score assigned to each correspondence measures how well the instrument meets the needs of the study: 1 being “barely addresses the need”; 2 being “partially addresses the need”; 3 being “fully addresses the need”.

**Table 5 ijerph-19-16050-t005:** Falls assessment tools scores.

	Used on Older Adults Living at Home	Validation	Number of Items	Specificity	Ease of Use	Usability by Different Professionals
Falls Efficacy Scale (FES) [75]	3	3	3	3	3	3
Modified Falls Efficacy Scale (MFES) [76]	3	3	3	3	3	3
Short Falls Efficacy Scale—International (FES-I) [77]	3	3	3	3	3	3
Survey of Activities and Fear of Falling in the Elderly (SAFE) [78]	3	3	2	3	3	3
Mobility Efficacy Scale (MES) [79]	3	2	3	2	3	3
University of Illinois at Chicago Fear of Falling Measure (UIC FFM) [80]	3	3	2	2	2	3
Fall Risk Index (FRI-21) [81]	3	3	3	3	3	3
Falls Risk Awareness Questionnaire (FRAQ) [82]	3	3	3	3	2	2
Fracture Risk Assessment Tool (FRAX) [83]	2	3	2	3	1	1
Berg Balance Scale [84]	3	3	2	3	1	1
Fullerton Advanced Balance (FAB) Scale [85]	3	3	2	3	1	1
Activities-specific Balance Confidence scale (ABC) [86]	3	3	3	2	3	3
Fall risk assessment tool (FRAT) [87]	2	3	2	3	1	1
Morse Fall Scale [88]	1	3	2	2	2	1
St. Thomas Risk Assessment Tool in Falling in Elderly Inpatients (STRATIFY) [89]	1	3	2	1	1	1
Conley Scale [90]	2	3	2	1	3	3
Johns Hopkins Fall Risk Assessment Tool (JHFRAT) [91]	3	3	3	2	3	3
Hendrich II Fall Risk Model [92]	2	3	1	2	1	1
Austin Health Falls Risk Screening Tool (AHFRST) [93]	2	2	1	2	2	1
Falls Risk for Hospitalized Older People (FRHOP) [94]	1	2	1	2	1	1
Falls Risk for Older Persons—Community Setting Screening Tool (FROP Com) [95]	3	3	2	1	3	3
Downton Fall Risk Index [96]	1	3	2	2	1	1
Elderly Fall Screening Test (EFST) Tool [97]	3	2	2	1	3	3
Home-Screen Scale [98]	3	2	2	2	3	3
Safety House Checklist [99]	3	1	2	2	3	3
Tinetti Performance-Oriented Mobility Assessment (POMA) [71]	2	3	2	3	1	1

The score assigned to each correspondence measures how well the instrument meets the needs of the study: 1 being “barely addresses the need”; 2 being “partially addresses the need”; 3 being “fully addresses the need”.

**Table 6 ijerph-19-16050-t006:** Cognitive decline assessment tools scores.

	Used on Older Adults Living at Home	Validation	Number of Items	Specificity	Ease of Use	Usability by Different Professionals
Mini Mental State Examination (MMSE) [100]	3	3	3	3	2	2
Modified Mini Mental State Examination (3MS) [101]	3	3	3	3	2	2
Montreal Cognitive Assessment (MoCA) [102]	3	3	3	3	2	2
Cognitive Abilities Screening Instrument (CASI) [103]	3	3	2	3	2	2
Short portable mental status questionnaire (SPMSQ or Pfeiffer) [104]	3	3	3	2	3	3
Abbreviated Mental Test (AMT) [105]	3	3	3	2	3	3
6-item cognitive impairment test (6-CIT) [106]	3	3	2	2	3	3
Clifton Assessment Procedures for the Elderly (CAPE) [107]	3	3	2	3	2	2
Clinical Dementia Rating scale (CDR) [108]	3	3	2	2	3	2
Controlled Oral Word Association Test (COWAT) [109]	3	3	2	2	1	2
Isaac Set Test (IST) [110]	3	3	2	2	1	2
Trail Making Test A & B (TMT) [111]	2	3	2	1	1	1
Rey Auditory Verbal Learning Test (RAVLT) [112]	2	3	2	1	1	1
Clock-drawing test (CDT) [113]	3	3	2	3	3	3
Brief Cognitive Screening Battery (BCSB) [114]	2	3	2	3	2	1
Mini-Cog [115]	3	3	3	2	3	3
Telephone Interview for Cognitive status (TICS) [116]	3	3	2	3	3	3
Cognitive screening instrument for dementia (CSID) [117]	3	3	2	3	3	3

The score assigned to each correspondence measures how well the instrument meets the needs of the study: 1 being “barely addresses the need”; 2 being “partially addresses the need”; 3 being “fully addresses the need”.

**Table 7 ijerph-19-16050-t007:** Loneliness assessment tools scores.

	Use on Older Adults Living at Home	Validation	Number of Items	Specificity	Ease of Use	Usability by Different Professionals
The UCLA Loneliness Scale [118]	3	3	3	3	2	2
The De Jong Gierveld scale [119]	3	3	3	3	2	2
Three-Item Loneliness Scale [120]	3	3	3	3	3	2
The Questionnaire to define Social Frailty Status (QSFS) [121]	3	1	3	2	3	3
The Social Frailty Index (SFI) [122]	3	1	3	2	2	3
The Steptoe Social Isolation Index [123]	3	3	3	3	2	3
The 11-items Duke Social Support Index [124]	3	3	3	3	2	2

The score assigned to each correspondence measures how well the instrument meets the needs of the study: 1 being “barely addresses the need”; 2 being “partially addresses the need”; 3 being “fully addresses the need”.

**Table 8 ijerph-19-16050-t008:** Support network assessment tools scores.

	Use on Older Adults Living at Home	Validation	Number of Items	Specificity	Ease of Use	Usability by Different Professionals
UCLA Loneliness Scale [119]	3	3	3	3	2	2
De Jong Gierveld scale [120]	3	3	3	3	2	2
Older Americans Resources and Services Program (OARS) Multidimensional Functional Assessment of Older Adults (MFAQ) [126]	3	3	2	3	3	3
Inventory of Social Supportive Behaviors (ISSB) [127]	3	2	1	3	3	3
Social Provision Scale [128]	3	3	3	3	3	3
Social support behaviors (SS-B) scale [129]	2	3	1	2	2	3
Medical Outcome Study Social Support Survey (MOS-SSS) [130]	3	3	3	3	3	3
Duke Social Support Index [137]	3	3	3	3	2	2
Multidimensional Scale of Perceived Social Support [131]	3	2	3	2	3	3
Lubben Social Network Scale [132]	3	2	3	3	3	3
Berkman–Syme Social Network Index [133]	3	3	3	3	3	3
Personal Resource Questionnaire (PRQ) [134]	2	3	3	3	3	3
Six social support deficits [135]	3	3	3	2	3	3
2-Way Social Support Scale [136]	1	3	2	3	3	3
Philadelphia Geriatric Center Morale Scale (PGCMS) [138]	2	3	2	3	2	3

The score assigned to each correspondence measures how well the instrument meets the needs of the study: 1 being “barely addresses the need”; 2 being “partially addresses the need”; 3 being “fully addresses the need”.

## Data Availability

Not applicable.

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
