# Peer review of "Assessment Tools of Biopsychosocial Frailty Dimensions in Community-Dwelling Older Adults: A Narrative Review"

_ijerph, 2022, doi:10.3390/ijerph192316050_

Round 1

Reviewer 1 Report

REVIEWER COMMENTS: Assessment tools for frailty multidimensional screening in community-dwelling older adults: a narrative review.

I would like to thank the authors for this work, as I feel it could make contributions to health care area. However, it needs major revisions to be published.

Frailty can be a physical or multidimensional construct, in the absence of consensus. But it is assumed as the sum of the parts, not as separate items. For example, if a questionnaire is used to assess loneliness, what we have measured is loneliness, not frailty. Actually a multidimensional tool measures multidimensionality, not one of its parts. In this sense, the title does not faithfully reproduce what has been analyzed in the article.

- Thus, extensive changes must be applied to the document. The title, the justification of the need for the work and the proposed objective must be reviewed. An approach such as risk factors could be a valid option.

In addition, other indications must be addressed:

Introduction section:

- The authors propose this objective of the article: “The objective of this narrative review is to identify more detailed and sensitive screening tools, to be coherently connected to each of the SUNFRAIL tool items, and further assess the corresponding compromised domains. Such more in-depth assessment is intended to develop adequate intervention strategies for frailty prevention dedicated to community-dwelling older persons, optimizing resources”. I wonder if there is a scale that already measures multidimensional frailty with 9 items, what novelty do you contribute to professionals aimed at speeding up frailty screening processes? The proposal is to use more scales? Another approach must be provided.

-  You refer to the SUNFRAIL scale, which structural validity has not been assessed and therefore we still do not have evidence about the possibility of combining the items proposed in the scale. Instead of focusing on the items of this scale, you should propose in the introduction a review of multidimensional scales such as the ones you refer to (for example the Edmonton Frailty Scale [13], the Tilburg Frailty Indicator [14], 77 and the Gerontopole Frailty Screening Tool, Kihon checklist...) and, based on this approach, you can propose your analysis (as you will see, many multidimensional scales share similar dimensions).

Materials and Methods section:

- Detail the criteria used to assess the sensitivity of the scales used (as you refer: “The objective of this narrative review is to identify more detailed and sensitive screening tools….”).

- Detail the criteria used (in a general way) to exclude articles on the basis of full text.

- Detail the criteria used to assess the specificity of the selected scales (as you refer: “specificity, as the ability of the tool to correctly screen for the risk factors of the specific ítem”.  

 Results:

 - SF-36 should not be considered within the scales on physical tests, it is a scale of perceived health.

Reviewer 2 Report

The authors conducted an extensive narrative review to summarize current evidence of tools to investigate frailty subdomains connected to previously indentified SUNFRAIL tool items. SUNFRAIL framework integrates the biomedical approach to frailty with the psycosocial one and generated a tool to identify frailty in community-dwelling older individuals. The authors searched for tools connected to SUNFRAIL subdomains in order to allow a early diagnosis of frailty.

The manuscript is well written and methodologically well performed; results are presented in a precise and detailed fashion, supported by tables and appropriate references. I enjoyed reading this manuscript which may serve as a important source for future research in the field. I have only some minor revisions regarding 2 points:

-Add legends to colored tables.

-Flow diagram of tools extracted for each subdomain is well described; however, I suggest reporting reasons for exclusion of manuscript after full-text evaluation in each figure.
